# Heart Rate Assessment during Neonatal Resuscitation

**DOI:** 10.3390/healthcare8010043

**Published:** 2020-02-23

**Authors:** Peter A. Johnson, Georg M. Schmölzer

**Affiliations:** 1Centre for the Studies of Asphyxia and Resuscitation, Neonatal Research Unit, Royal Alexandra Hospital, Edmonton, AB T5A 0A1, Canada; paj1@ualberta.ca; 2Department of Pediatrics, University of Alberta, Edmonton, AB T5A 0A1, Canada

**Keywords:** infants, newborn, neonatal resuscitation, heart rate

## Abstract

Approximately 10% of newborn infants require some form of respiratory support to successfully complete the fetal-to-neonatal transition. Heart rate (HR) determination is essential at birth to assess a newborn’s wellbeing. Not only is it the most sensitive indicator to guide interventions during neonatal resuscitation, it is also valuable for assessing the infant’s clinical status. As such, HR assessment is a key step at birth and throughout resuscitation, according to recommendations by the Neonatal Resuscitation Program algorithm. It is essential that HR is accurate, reliable, and fast to ensure interventions are delivered without delay and not prolonged. Ineffective HR assessment significantly increases the risk of hypoxic injury and infant mortality. The aims of this review are to summarize current practice, recommended techniques, novel technologies, and considerations for HR assessment during neonatal resuscitation at birth.

## 1. Introduction

The transition from intrauterine to extrauterine life relies on major physiological changes at birth [1,2]. These changes include clearing of liquid from the lung to allow entry of air into distal gas-exchange regions, relaxation of the pulmonary vascular bed to increase pulmonary blood flow, increases in systemic vascular resistance, cessation of umbilical venous return, occlusion of the fetal shunts, and oxygen saturation, and increases in heart rate (HR) [1,3]. Although most infants make this transition from placental to pulmonary gas exchange at birth without help, ~10% of newborn infants require positive pressure ventilation (PPV) assistance and ~1% of these infants require advanced resuscitative interventions including chest compressions and medications [2].

International neonatal guidelines including the Neonatal Resuscitation Program (NRP) use predefined HR targets of 100 and 60 beats per minute (bpm) to initiate PPV and chest compressions, respectively [1,2,4]. These cutoffs are chosen arbitrarily, as neither human nor animal data are available to support these cutoffs [5]. Published normograms have reported the median HR at 1 minute can be <100 bpm in newborn infants that do not require any medical intervention [6]. At 2 minutes of age, HR increases to >100 bpm, with the increase in HR being slower in preterm infants and infants born by caesarian section [6]. Ultimately, it is crucial to recognize the successfulness of resuscitations rely largely on how effectively and rapidly the appropriate interventions are delivered based on changes in HR, as opposed to responding according to pre-established cutoffs. 

Espinoza et al. used a post-transition asphyxia-induced bradycardia term piglet model and observed an increase in HR >100 bpm in only 6/30 (20%) piglets after 30 s of adequate PPV [7]. The current NRP recommendation contrasts this observation, stating that 15 s of PPV at birth should result in an increase in HR [4]. In term apneic newborn infants, an adequate tidal volume will result in increased HR [8]. A similar relationship was observed in preterm infants during PPV and during sustained inflation [9,10,11,12]. Furthermore, in infants <30 weeks’ gestation, an increase in HR >100 bpm required a median (IQR) times of 73 (24–165) s of PPV [13]. Additionally, HR only stabilized once HR was >120 bpm, requiring a longer median (IQR) time of 243 (191–351) s of PPV [13]. For infants >34 weeks gestation, it was determined that a decrease in HR to <100 bpm from a pause in PPV results in a 2-fold increase in the risk of death, whereas an increase to >100 bpm during the initial treatment improves survival by 75% [14]. 

Therefore, if HR is detected too slowly or inaccurately, it will delay critical interventions or lead to inappropriate interventions, which are ineffective to improve the infant’s status and/or increase the risk of cardiac arrest. 

Current NRP guidelines recommend the use of umbilical cord palpation, auscultation, pulse oximetry (PO), and electrocardiography (ECG) for HR assessment during neonatal resuscitation at birth [1,2,4,15]. However, there have been some concerns about their accuracy, latency or feasibility for clinical assessment [16,17,18,19,20]. Below, we discuss recommended HR assessment approaches, their outcomes, and ongoing challenges (Table 1).

## 2. Auscultation/Palpation

Palpation involves the assessment of a pulse at the umbilical, femoral, or brachial arteries, whereas auscultation involves using a stethoscope to listen to heart beats, normally from the chest of the infant [20,21]. The NRP recommends counting the heart beats heard over 6 s and multiplying by 10 to determine HR in bpm [22]. Accounting for placement, pulse detection, listening window of 6 s, and time required for mental computation, this technique allows for quick approximation of HR. A total HR assessment time ranging from 7–19 s on average have been previously reported for both palpation and auscultation [16,23,24].

Owen & Wyllie compared palpation at the femoral and brachial artery and umbilical cord in newborn infants to assess the accuracy of calculating a HR >100 bpm [21]. Auscultation using a stethoscope provided HR >100 bpm in 100% of the cases, whereas palpation did not always result in a palpable HR [21]. Palpation of the umbilical pulse was accurate for 55% of cases, compared to 20% and 25% at femoral and brachial pulse, respectively [21]. Moreover, a concerning 25% and 60% of participants were unable to palpate a pulse, while 15% and 45% incorrectly assessed HR as <100 bpm using the femoral and brachial pulse, respectively [21]. Therefore, auscultation is more accurate than palpating from any of the three locations, but when a stethoscope is not available, palpation of the umbilical cord provides greater accuracy.

Chitkara et al. and Boon et al. randomized healthcare providers to either auscultation or palpation, blinding them to high-fidelity simulated neonatal resuscitation scenarios [22,25]. Healthcare providers were randomized to scenarios representing the NRP HR target ranges at >100, 60–100, <60 bpm and required to perform an initial assessment followed by subsequent assessments. Both studies reported the greatest accuracy of HR at <60 bpm, followed by 60–100 bpm, and then >100 bpm [22,25]. Chitkara et al. additionally determined no difference between initial and subsequent assessments, with errors occurring an alarming 26–48% and 26–52% of the time, respectively. However, a more recent simulation study by Money et al. evaluated the accuracy of auscultation according to NRP HR target ranges and identified overestimation of HR <60 bpm and underestimation of HR >100 bpm as a common tendency for participants [26]. The latter observation was similar to a study by Kamlin et al., who compared auscultation and umbilical cord palpation with ECG in term newborn infants and reported both auscultation and umbilical cord palpation underestimated HR with a mean HR difference of 14 and 21 bpm, compared to ECG [27]. These studies suggest HR assessment using palpation or auscultation are inaccurate, thereby resulting in a greater number of incorrect assessments for determining HR at birth. This is concerning, as under- or overestimation of HR can result in inappropriate management (i.e., early or delayed interventions) in 28% of cases in a simulated environment alone [16]. During neonatal resuscitation in the delivery room, assessment of HR using auscultation remains challenging. Resuscitators need to assess HR while working under high stress levels, high cognitive load, and varying surrounding noise levels. They further have to pay attention and focus on the task, which is of particular importance within the first few minutes after birth as a wide HR variability might occur and those responses are unpredictable compared to manikins.

## 3. Pulse Oximetry

PO can provide a continuous and simultaneous measure of SpO_2_ and HR from the infants hand, wrist, or foot [28,29]. It functions by using two light diodes, which emit light at red and infrared frequencies, and a photo-detector that measures changes in the transmitted light from the oxygenated and deoxygenated blood to determine oxygen saturation [30]. HR can be determined using the light intensity changes, which corresponds to pulsatile blood volume changes in the artery [30]. International guidelines recommend the use of PO during PPV or when providing supplemental oxygen [1,2]. However, there are several limitations of PO to monitor HR including (i) delays in time needed to display first HR values [31,32], (ii) potential underestimation of HR compared to ECG outcomes [19], and (iii) difficulties in obtaining a good signal quality when HR <100 bpm [19,31]. Other limitations include: low peripheral perfusion, the effect of transitional circulation, low volume state, vernix effects, skin oedema, acrocyanosis, signal dropout, movement artefacts, arrhythmias, and presence of ambient lighting, which might delay or interfere with PO HR measurements [33,34,35,36,37].

The vast majority of studies examining the accuracy and reliability of PO for HR assessment utilize ECG for comparison [15]. We have identified six studies comparing PO to ECG for HR assessment in the delivery room [19,23,31,32,38,39] and one in the neonatal intensive care unit (NICU) [40]. While accuracy is most commonly described as the level of association with the gold standard (ECG for most cases), reliability is defined by detection and signal quality of a waveform (PO or ECG). Kamlin et al. analyzed 5877 data pairs of ECG HR and good-quality PO HR (defined by the presence of signal bars and no “low-signal quality” message) in 55 preterm or term infants reporting a mean (2 SD) difference between ECG HR and PO HR as −2 (26) bpm overall and −0.5 (16) bpm in infants who received either positive-pressure ventilation and/or cardiac massage [31]. However, at ECG HR <100 bpm, good-quality PO HR <100 bpm could only be detected 89% of the time [31]. While these former results suggest a strong accuracy for PO HR monitoring at birth when compared to ECG HR monitoring, the latter suggests the need to explore specific outcomes during bradycardia. In a study by Iglesias et al., both PO and ECG were used to detect bradycardia (HR <100 bpm) during stabilization [38]. PO detects both the start and end of bradycardia episodes a median time of 5 seconds slower than ECG [38], which is concerning as it could lead to delayed initiation of resuscitation interventions or the unnecessary prolongation of interventions. A study by van Vonderen et al. examined the accuracy of PO, compared to ECG, for HR assessment in the first minutes after birth [19]. PO underestimated HR, displaying HR <100 bpm and suggesting bradycardia in the first minutes after birth in uncompromised infants [19]. This underestimation was verified by the weaker association of PO HR with left ventricular outflow when compared to ECG HR, which suggests PO missed beats and is unreliable for detecting all pulse waves from the peripheral vasculature in the immediate transition [19]. With the low accuracy and reliability of PO during the first minutes of life, including the Golden Minute when HR assessments are strongly recommended, the latency of signal detection must be a major consideration for HR assessment at birth.

Unfortunately, long latency ranging from 1–2 min for sensor attachment and reliable signal display following birth are reported for PO, indicating HR is not detected within the Golden Minute [15,41]. In a study by Mizumoto et al., achieving a reliable PO and ECG signal at birth required a median (IQR) time of 122 (101–146) vs. 38 (34–43) s, respectively [32]. Furthermore, HR detection was more difficult using PO when compared to ECG, in bradycardic newborn infants with poor perfusion [32]. Therefore, healthcare providers must not rely exclusively on PO, especially during bradycardia, as PO might underestimate HR. Two further studies have examined the fastest approach to detect HR using two different PO application techniques [18,42]. One technique involves attaching the PO sensor to the oximeter first (STOF), whereas the other requires the attachment of the sensor to the infant first (STIF). While the first study determined the STIF method was faster and more reliable providing data within 90 s after birth [18], the second showed suggested STOF had a faster signal acquisition time, although both techniques provided a similar time from birth to a reliable signal [37]. In spite of these limitations, PO is valuable for HR assessment in various special cases in the delivery room where ECG may not be effective.

## 4. Electrocardiography

ECGs used for neonatal resuscitation utilizes three-electrode ECGs, which typically involves the placement of these electrodes on right arm, left arm and left leg or abdomen [42]. Electrical activity originating from the sinoatrial node of the heart is recorded by these electrodes and used to generate a continuous ECG waveform. As ECGs utilize R-wave detector algorithms using the QRS segment for calculation of HR, continuous ECG waveforms containing the QRS complex can be utilized to confirm the reliability of the signal [42,43]. ECG is currently considered the “gold standard” for HR measurements, which has been demonstrated to be accurate and reliable in displaying HR compared to the above mentioned recommended techniques [1,44]. 

We have identified three randomized clinical trials assessing latency, reliability or neonatal resuscitation outcomes when using ECG for HR assessment through a systematic review [13,20,36,37]. While Murphy et al. performed two trials in low-risk infants [23,39], Katheria et al. conducted a trial in preterm infants during stabilization at birth [40]. All studies involved blinding and randomized allocation of the infants into a PO or ECG group. Moreover, all studies reported a faster median (IQR) time for HR assessment using ECG compared to PO at birth [24 (19–39) s versus 48 (36–69) s and 66 (46–86) s versus 114 (75–153) s]. This suggests that HR detection during the Golden Minute may or may not be achieved by ECG. In a recent study by Gulati et al., a novel technique has been suggested to overcome this delay by pre-setting ECG electrodes on the bed for attachment to the infants back, while also allowing for chest compressions not to interfere with ECG signals [45]. However, this technique has yet to be evaluated in infants requiring resuscitation. 

Additionally, Murphy et al. determined auscultation and PO underestimated ECG HR by a mean difference (95% confidence interval) of −9 (−15 to −2) and −5 (−12 to 2) bpm, respectively [23]. This supports previous studies, which identified the other techniques underestimate ECG HR for infants with HR >100 bpm. In the trial by Katheria et al., it was determined PO HR was lower than ECG HR in the first two minutes of life, yet no significant differences were determined in time to the delivery of the appropriate interventions in both groups [40]. While this is encouraging, this initial underestimation may be more critical in high-risk infants that require advanced interventions such as chest compression. In fact, a recent retrospective study suggests ECG use is associated with increasing administration of chest compressions and fewer endotracheal intubations in the delivery room [46]. Another benefit of having an early, reliable HR is the improved preparedness of the clinical team for any intervention.

Despite this efficacy, the use of ECG also has several limitations including (i) time needed to clean newborns’ skin (e.g., from blood, vernix, mucus or amniotic fluid), (ii) potential signal interference from suboptimal placed ECG-lead (due to aforementioned fluids), (iii) potential damage and risk of infection caused by the leads on the delicate skin of premature infants, and (iv) or special clinical cases involving hydrops fetalis [17] or pulseless electric activity (PEA) which could result in misinterpretation of displayed ECG HR [24,42,47].

### Pulseless Electrical Activity

PEA is a phenomenon that occurs when cardiac output is zero but the ECG still displays an HR. PEA involves cardiac electrical activity in the absence of a detectable pulse and has been reported in adults and children, most commonly after hypoxia, severe volume loss, sepsis, tension pneumothorax or following cardiac arrest [7]. There is increasing evidence that PEA occurs in the delivery room [15]. Two studies report that the ECG displayed a HR during PEA in 40–50% of asphyxiated newborn piglets [48,49]. There have been one case report and a case series totaling seven cases of PEA in the delivery room during neonatal resuscitation [50,51,52,53]. This is concerning, especially if healthcare professionals are relying exclusively on the ECG signal. For current practice, we therefore recommend assessments using a combination of current recommendations: palpation, auscultation, PO and ECG, in light of these cases.

## 5. Novel Technologies

There are several emerging HR assessment technologies, which have been recently summarized in There are several emerging HR assessment technologies, which have been summarized over the recent years in three systematic reviews (Table 2; Figure 1A,B) [15,41,54]. These novel technologies can be classified as contact (i.e., ECG, PO, dry-electrode ECG, electrical velocimetry, reflectance photoplethysmography, electromyography), intermittent contact (i.e., auscultation/palpation, Doppler ultrasound, digital stethoscope), non-contact or sensor-based (i.e., camera-based photoplethysmography, capacitive sensors, piezoelectric sensors, laser Doppler vibrometry), and assistive technologies (i.e., tap-based smartphone apps). While contact technologies require continuous contact with the infant and may be adverse for the fragile, premature infant skin, intermittent contact is less intrusive whereas non-contact methods are the least intrusive. Updates to existing techniques, such as the use of the digital stethoscope and smartphone apps to enhance accuracy and time required for auscultation [15], and application of electrodes on the infant’s back for improved ECG latency and reliability during resuscitation, have also been identified. While these technologies are promising, further evaluation is needed before they can be translated into routine use.

## 6. Low Resource Settings

Despite international guidelines, there exist several differences in neonatal resuscitation practices at birth in different regions around the world. Technologies such as ECG and PO are costly and inaccessible in low-resource settings. In these settings, auscultation and palpation are most suitable and often the only approach. While it is possible to obtain continuous HR data by dedicating one member of the clinical team to continuous HR assessment via auscultation or palpation, this places a higher demand on the number of members required on the team [55]. In addition to environmental factors such as the availability of resources, the availability of clinical staff, their levels of training, and competency of healthcare professionals may also influence the success of the resuscitation. 

## 7. Conclusions

Heart rate assessment is vital at birth to guide neonatal resuscitation. However, current neonatal resuscitation guidelines recommend the use of auscultation/palpation, pulse oximetry, and electrocardiography for heart rate assessment. While auscultation/palpation are fast and reliable, they are inaccurate in some instances. Pulse oximetry and electrocardiography are superior in accuracy compared to auscultation/palpation, however they require a longer time to assess the initial heart rate. 

## Figures and Tables

**Figure 1 healthcare-08-00043-f001:**
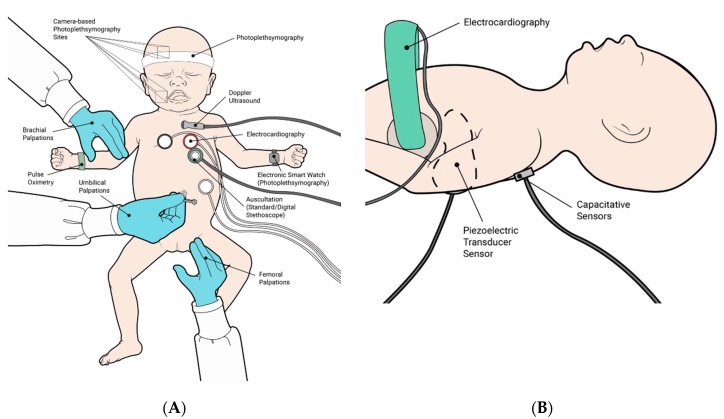
Current and novel technologies or techniques for heart rate assessment identified by Johnson et al. [15]. (**A**): An illustration of current recommendations, including electrocardiography, pulse oximetry, auscultation with a standard stethoscope, and palpation from brachial, femoral, and umbilical arteries, and novel technologies, including auscultation with a digital stethoscope, Doppler ultrasound, photoplethysmography, camera-based photoplethysmography for heart rate assessment. (**B**): An illustration of novel technologies including dry-electrode electrocardiography, and two sensor-based methods for heart rate assessment. Reproduced with permission from RETAIN Labs Medical Inc. (https://www.playretain.com).

**Table 1 healthcare-08-00043-t001:** Recommended techniques for heart rate assessment, associated outcomes including accuracy, time required for assessment, and reliability of technique, and limitations. Abbreviations: HR: heart rate; PO: pulse oximetry ECG: electrocardiography; bpm: beats per minute; sec: seconds.

HR Assessment Technique	Palpation	Auscultation	PO	ECG
Accuracy	Underestimates HR >100 bpm, fairly accurate <100 bpm Underestimates 21 bpm, compared to ECG	Underestimates HR >100 bpm, fairly accurate <100 bpm One study suggests healthcare providers overestimate HR <60 bpm	Accurate, but underestimates HR in the first minutes of life (~2 min)	“Gold standard”
Time required for assessment	~7–19 sec	~7–19 sec	~60–120 sec	~30–60 sec
Method to confirm reliability	Feeling pulse	Hearing heartbeats	Observing a regular waves on the PO waveform	Observing regular QRS complexes on ECG waveform
Limitations	Requires great deal of concentration and attention Factors such as noise, cognitive load, and stress can result in inaccurate HR	Requires great deal of concentration and attention Factors such as noise, cognitive load, and stress can result in inaccurate HR	High latency for reliable HR detection (48 sec from sensor application) Underestimates HR in first 2 min Low peripheral perfusion, volume, movement, ambient lighting, etc. can result in loss or unreliable HR signal	High latency (24 sec from lead application) Requires time for cleaning of skin from fluids Increases risk of skin damage, injury, or infection in premature infants PEA, hydrops & other special cases may result in loss or unreliable HR signal

**Table 2 healthcare-08-00043-t002:** Novel technologies or techniques for heart rate assessment identified in this review and three systematic reviews [15,41,54]. Each technology is characterized according to its classification, functionalities and strengths, and limitations. Abbreviations: HR: heart rate; ECG: electrocardiography; PO: pulse oximetry; PPG: photoplethysmography; EMG: electromyography; cPPG: camera-based photoplethysmography; STIF: sensor to infant first method; STOF: sensor to oximeter first method; sec: second.

Novel Technology/Technique	Classification	Description	Limitations
ECG presetting	Contact	ECG sensors can be preset in a triangle formation facing up on the bed easier & faster HR detection upon delivery & ease of access during chest compressions.	HR signal loss is more frequent than conventional method.
PO sensor application techniques	Contact	Either STIF or STOF methods may result in faster HR detection time. Studies report similar reliability for both techniques.	Uncertainty which technique is faster at birth.
Dry-electrode ECG	Contact	Uses dry-electrodes to detect reliable HR with short latency (within ~7–8 sec)	Requires drying infant prior to use & movement causes interference.
Electrical velocimetry	Contact	Uses blood conductivity to measure cardiac output, stroke volume, & HR, providing accurate HR compared to ECG.	Only assessed in term infants before & movement causes interference.
Reflectance PPG/PO	Contact	Uses reflectance instead of transmission to monitor SpO_2_ & HR, providing accurate HR.	Similar limitations to PO.
Transcutaneous EMG	Contact	Uses electrical activity of muscle tissue & has a high degree of accuracy compared to ECG	Similar to ECG and have limited advantages over it.
Doppler ultrasound	Contact, intermittent contact	Uses ultrasound frequency sound waves to detect HR accurately & within a short period of time.	Movement can affect skin-gel interface, while noise & ventilation can interfere with audible & visual signal, respectively.
Digital stethoscope	Intermittent contact	Uses electronics to augment sound detected by auscultation with greater clarity to improve HR accuracy.	Influenced by movement & noise & has similar limitations to auscultation.
cPPG	Non-contact	Uses changes in wavelengths over a region of interest to determine HR, offering a high degree of accuracy to ECG.	Signal loss is common about 20% of the time due to ambient light, movement, & obstructions.
Capacitive sensors	Non-contact, sensor-based	Forms a capacitive electrode between the infants’ skin & an electrode without directly touching the infant to determine an accurate ECG signal.	Signal loss is common about 15% of the time due to movement, etc.
Piezoelectric sensors	Non-contact, sensor-based	Uses acoustic vibrations from heartbeats to produce electrical signals providing HR, offering accurate data compared to ECG.	Movement from ventilation, infant movement, or resuscitator movement greatly affects signals.
Laser Doppler	Non-contact	Uses a laser beam to detect movements in thoracic walls of infant due to cardiac activity, providing a fairly accurate HR compared to ECG.	There is uncertainty as well as a high cost & complexity associated with the system.
Tap-based smartphone apps	Assistive	Uses screen tapping, which is paired with auscultation to detect HR based on timing between heartbeats and provides a fast and accurate HR in simulation scenarios. Also useful and accessible in low-resource settings.	Technical software problems, risk of infection with smartphone use, requires auscultation & therefore has the same limitations

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
