# Peer review of "Heart Rate Assessment during Neonatal Resuscitation"

_healthcare, 2020, doi:10.3390/healthcare8010043_

Round 1

Reviewer 1 Report

This paper provides a very focused view on heart rate monitoring assessment at delivery and is a valuable overview on this topic. I don’t have major concerns nor comments, only one suggestion. The authors mainly discuss aspects related to HR as absolute, rather dichotomous variables. Is there any new information on heart rate variability assessment and subsequent outcome (so more on HR pattern analysis besides HR acquisition ?

Some other questions to reconsider the text:

What do you or the authors mean with stabilized HR (line 45) [Stabilized once HR was >120 bpm…] Same line, a rapid decrease = if I get this correct, it is a rapid return to bradycardia ?

Table 1: can you also ‘quantify’ the latencies mentioned ?

Line 103: iii) difficulties of obtaining HR values <100 bpm due to poor signal.: to what extent is the poor signal detection limited or specific to < 100 bpm settings ?

Table 2: please check the legend, the text is incomplete.

Minor

The reference list needs additional attention and verification, ref 1,17,22,26,27,32,43,59 are for sure incomplete.

Line 38: suggest to rephrase, Espinoza et al, used a term piglet using ?

Author Response

Thank you for your comments.

We have used your comments to improve the presentation of the manuscript.

Sincreley

Georg Schmolzer

Reviewer 2 Report

The rationale of the review seems clear and there are useful practice points in it. Presenting the review in the four methods of HR assessment is logical.

However, the floe of ideas in each section needs to be improved and extensive review of the English language used is recommended. Some sentences read as verbatim extracts from the references. Examples of unclear sentences or those in which there are gross grammatical errors include: Lines 34,33,36,41,42,62,68,70,71,72, and 90.

Author Response

Thank you for your comments.

We have used your comments to improve the presentation of the manuscript.

Sincerely

Georg Schmolzer

Round 2

Reviewer 2 Report

I am impressed by the inputs in the revised version of the manuscript. The paper is easily readable, the message is clearly communicated. The findings and conclusion seems valid and will help advance the practice around neonatal resuscitation, a crucial intervention to improve newborn survival. 

Thank you.